# DYNAMICS-AWARE SKILL GENERATION FROM BEHAVIOURALLY DIVERSE DEMONSTRATIONS

## ABSTRACT

Learning from demonstrations (LfD) provides a data-efficient way for a robot to learn a task by observing humans performing the task, without the need for an explicit reward function. However, in many real-world scenarios (*e.g.*, driving a car) humans often perform the same task in different ways, motivated not only by the primary objective of the task (*e.g.*, reaching the destination safely) but also by their individual preferences (*e.g.*, different driving styles), leading to a multi-modal distribution of demonstrations. In this work, we consider a Learning from state-only Demonstration setup, where the reward function for the common objective of the task is known to the learning agent; however, the individual preferences leading to the variations in the demonstrations are unknown. We introduce an imitation-guided Reinforcement Learning (RL) framework that formulates the policy optimisation as a constrained RL problem to learn a diverse set of policies to perform the task with different constraints imposed by the preferences. Then we propose an algorithm called LfBD and show that we can build a parameterised solution space that captures different behaviour patterns from the demonstrations. In this solution space, a set of policies can be learned to produce behaviours that not only capture the modes but also go beyond the provided demonstrations.

## 1 INTRODUCTION

Learning from demonstrations (LfD) (Schaal, 1996) provides an alternative way to Reinforcement Learning (RL) for an agent to learn a policy by observing how humans perform similar tasks. However, in many real-world scenarios (*e.g.*, driving a car), humans often perform the same task in different ways. Their behaviours are not only influenced by the primary objective of the task (*e.g.*, reaching the destination safely) but also by their individual preferences or expertise (*e.g.*, different driving styles) (Fürnkranz & Hüllermeier, 2010; Babes et al., 2011). In other words, all the underlying policies maximize the same task reward, but under different constraints imposed by individual preferences. This leads to a multi-modal distribution of demonstrations, where each mode represents a unique behaviour.

With multi-modal demonstrations, typical LfD methods, such as Behaviour Cloning and Generative Adversarial Imitation Learning, either learn a policy that converges to one of the modes resulting in a *mode-seeking* behaviour or exhibit a *mean-seeking* behaviour by trying to average across different modes (Ke et al., 2020; Ghasemipour et al., 2020; Zhang et al., 2020). While the former will still recover a subset of solutions, the latter may cause unknown behaviour (see Fig 1). Furthermore, none of these approaches is able to learn policies that correspond to the behaviours of a wide range of individuals. The problem becomes even more challenging when there are only "state observations" in the demonstrations without the actions. In such situations, supervised or unsupervised learning approaches cannot be applied directly to find a policy, and the agent must interact with the environment or with a simulator (Torabi et al., 2019).

Being able to learn a diverse set of policies from demonstrations is often desirable to serve the requirements of a wide range of individual users. For instance, every self-driving car can have a driving policy (selected from the diverse set of pre-trained policies) that matches the preferences of a user. Many recent works show the advantages of having a diverse set of policies, for instance, rapid damage adaptation in robotics (Kaushik et al., 2020; Chatzilygeroudis et al., 2018; Cully et al., 2015) and safe sim-to-real policy transfer in robotics (Kaushik et al., 2022).

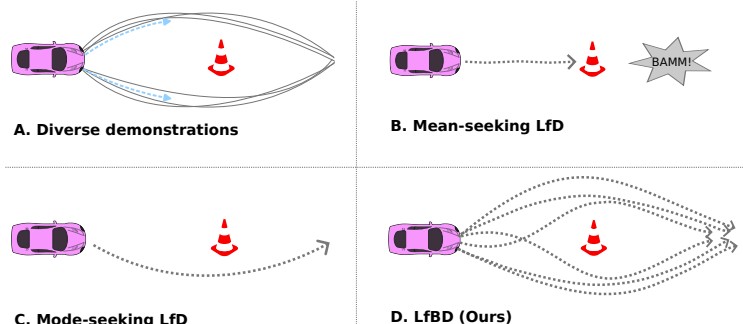

Figure 1: Given the demonstrations from several individuals in **A**, the mean-seeking policy produces unseen behaviour that is unsafe as shown in **B**, the mode-seeking policy only recovers one mode as shown in **C**. We propose a new framework that recovers all the possible solution modes as shown in **D**. The example is inspired from (Ke et al., 2020)

.

In this work, we consider a specific setup of LfD, known as *Imitation Learning from Observations alone* (ILfO) (Sun et al., 2019), where the learning agent only has access to the state observations without their corresponding actions. We propose a new framework that combines Reinforcement Learning with ILfO to solve the issues of learning from the multi-modal states-only demonstrations, especially with a small set of unlabelled demonstrations. Unlike most of the LfD methods, our goal is not just to learn *how to perform a task* or *how to mimic humans*, but rather *how to perform a task in all possible ways* as shown in Fig 1D. Thus, we focus on applications where a high-level task reward function can be defined easily, but the preference components that cause diverse behaviours cannot be explicitly defined. These include tasks such as autonomous driving or robotic manipulations, where the agent needs to mimic the human's behaviour pattern (*i.e.*, preference) while reaching the intended goal (*i.e.*, task reward).

In practice, defining a high-level task reward can be straightforward, i.e., for autonomous driving, this can be a function of the distance to a target location and the penalty for collisions. However, defining the preference component is far from easy, as it may be impossible to find a mathematical expression for each individual's preferences. Thus, we formulate the multimodal policy generation guided through demonstration as a constrained optimisation problem; where the generation of multimodal behaviours results from optimising policies for a given task reward function that satisfy different preference constraints.

As contributions, we first propose a new imitation-guided RL framework and an algorithm called Learning from Behaviourally diverse Demosntration (LfBD) to solve the problem of policy generation from multimodal (state-only) demonstrations. We then propose a novel projection function that captures preferences as state-region visitations. This projection function allows us to build a parameterised solution space to allocate policies such that they satisfy different preference constraints. We show that our approach is capable of generating multimodal solutions beyond the provided demonstrations, *i.e.* the resulting solutions also include interpolations between the provided demonstrations. Furthermore, our method allows us to perform different types of *post-hoc* policy searches in the solution space: 1) Given a (sate-only) demonstration, find the closest policy capable of generating this demonstration. 2) Search policies in the solution space that have a high/low likelihood according to the provided demonstrations (*i.e.*, similar to the provided demonstrations). 3) Find solutions that satisfy different constraints.

## 2 BACKGROUND

### 2.1 IMITATION LEARNING AS DIVERGENCE MINIMISATION

In this section, we discuss why current Imitation Learning (IL) methods are incapable of dealing with multi-modal demonstration distributions; especially, when only a small set of demonstrations is available. Zhang et al. (2020); Ke et al. (2020); Ghasemipour et al. (2020) have shown that current the Imitation learning (IL) methods can be derived as a family of *f-divergence* minimisation methods, where the divergence of the state-action distributions of the expert $p_{\pi_{exp}}(s, a)$ and learning

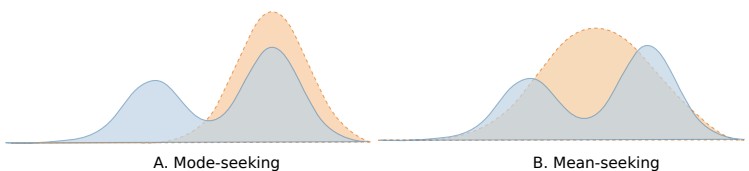

Figure 2: Fitting a Gaussian to a multimodal distribution (blue) by minimising the reverse-KL (A, mode-seeking) and the forward-KL (B, mean-seeking)

agent $p_{\pi_\theta}(s, a)$ is minimised using either Kullback-Lebler (KL) divergences or Jensen-Shanon (JS) divergence. As KL divergence is not symmetric, both forward KL and reverse KL are used. And depending on the choice of the divergence, the optimisation yields different results (Nowozin et al., 2016; Ke et al., 2020; Ghasemipour et al., 2020) (see Fig 2). Additional details in Appx. A.2.

*Behaviour cloning* (Pomerleau, 1988), a supervised off-line learning approach, is one of the examples that use forward KL to optimise its policy. In practice, it either follows a mean-seeking behaviour under a stochastic policy and assumes data sampled during training are balanced or exhibits a mode-seeking pattern. As it is a supervised learning method that lacks interaction with the environment, it often requires a large amount of demonstration data in order to accurately estimate the true state-action distribution (Neu & Szepesvári, 2009).

One example of using reverse KL is *Adversarial Inverse Reinforcement Learning* (Fu et al., 2017), which recovers both the reward function and the policy by formulating IL as a min-max optimisation using *Generative Adversarial Networks* (GAN) (Goodfellow et al., 2014). Here, the generator serves as the policy network and a discriminator as the reward function. The optimised objective is the reverse KL divergence, which can be expressed in terms of entropy $\mathcal{H}$ as: $\mathbb{E}_{p_{\pi_\theta}(s,a)} \left[ \log \pi_{exp}(s, a) \right] - \mathcal{H}(p_{\pi_\theta}(s, a))$. In practice, it faces additional challenges, such as the ambiguity of the reward function (Ng et al., 2000; Arora & Doshi, 2021) (i.e., many reward functions could generate the same behaviours). Thus, its policy learning counterpart, *Generative Adversarial Imitation Learning* (GAIL) (Ho & Ermon, 2016) is more frequently used. This latter optimises the JS divergence (Goodfellow et al., 2014; Ghasemipour et al., 2020; Ke et al., 2020) with the causal entropy as policy regularisation term.

Optimising the JS-divergence result in the same mode-seeking behaviour as the reverse KL (Theis et al., 2015). It further faces one of the most notorious problems of the GAN-based approach, commonly known as *mode collapse* (Srivastava et al., 2017; Bau et al., 2019), where the generator collapses to produce only a small set of data samples (partial collapse) or even a single sample (complete collapse), which disregards the multi-modal nature of the distribution. Furthermore, the adversarial optimisation scheme in GAN-based methods further imposes other challenges, such as instability in the optimisation. It is often difficult to guarantee that both the generator and the discriminator converge optimally at the same time (Radford et al., 2015). It is common to observe that the loss of the discriminator converges very quickly to zero (Arjovsky & Bottou, 2017; Xu et al., 2018) that impedes the gradient updates of the generator. It often needs a considerably large number of data and fine-tuning or additional regularization in order to stabilise the training process. The optimisation of IL methods is convex if the actions are deterministic and the demonstrations cover the complete state space (Neu & Szepesvári, 2009). In practice, it is always difficult to make sure that the demonstrations span the entire state space; instead, they just cover one of the global optima which ensures the convergence of the solution.

## 2.2 IMITATION LEARNING FROM OBSERVATION ONLY

Towards ILfO, Torabi et al. (2018) proposes a new approach called BCO, where an inverse dynamics model is learnt through interaction with the environment and a policy is learnt from state observations by using the action inferred by the inverse dynamics. However, as with most LfD methods, BCO does not handle multimodal demonstrations. Similarly, another approach called ILPO (Edwards et al., 2019) learns a forward dynamics model and a latent policy to circumvent the need for the actions; however, it is limited to discrete action space. Latent policy learning from demonstrations has also been proposed by Ren et al. (2020); Fujiishi et al. (2021); Wang et al. (2022a). GAIfO (Torabi et al., 2019) extends GAIL to state-only observations by using a discriminator that discriminates the state transitions instead of state-action pairs from the expert. *State-alignment*

*based Imitation Learning* (Liu et al., 2019) uses state alignment to recover state sequences close to the demonstrations using local state alignment and global state alignment. While it shares some similarities with our method (*i.e.*, state distribution matching), it needs state-action pairs to pre-training their model. A similar work is proposed in Lee et al. (2019). InfoGAIL (Li et al., 2017) combines InfoGAN (Chen et al., 2016) with GAIL to capture the variations of diverse demonstrations using interpretable latent factors. However, it is not a state-only approach and suffers, in addition, from the mode-collapse problem of GAN. (Gavenski et al., 2020) shares similarities with us by using the sampling-based method to imitate unknown policies but with state-action demonstrations. The closest work is perhaps by Ghasemipour et al. (2020), where they recover a multi-modal policy from state distributions that preserves 2 solution modes using a modified reverse KL. However, this multimodality is due to the use of the stochastic policy(*i.e.*, outputs mean and variance of a Gaussian distribution). Unlike our approach, it does not allow the *post-hoc policy search* we propose, therefore, it cannot obtain a specific policy deterministically. Shafiullah et al. (2022) uses a transformer to learn multimodal demonstrations; however, it does not deal with state-only data. Similar examples include the diffusion model (Wang et al., 2022b) for path planning that focuses on in-distribution data generation with access to the action set.

## 3 PROBLEM SETUP

Consider a finite-horizon Markov Decision Process (MDP) $\langle \mathcal{S}, \mathcal{A}, p, p_0, R, \gamma, T \rangle$, where $\mathcal{S} \subseteq \mathbb{R}^{d_s}$ and $\mathcal{A} \subseteq \mathbb{R}^{d_a}$ are the continuous *state* and *action spaces*, $p(s'|s, a)$ is the state-transition probability, $s$ and $s'$ are the current and the next state, $a$ is the applied action, $p_0$ the initial state distribution, $\gamma \in [0, 1]$ the *discount factor*, $R : \mathcal{S} \times \mathcal{A} \rightarrow \mathbb{R}$ the *reward function*, and $T$ the *time horizon*. We assume that this MPD has differentiable dynamics with a continuous states transition function $s_{t+1} = f_c(s_t, a_t)$, such that $d(s_t, s_{t+1}) < \delta$, with $d$ as a distance metric, and $\delta$ a small constant.

Let $\mathbb{D} = \{\tau_0, \tau_1, \ldots, \tau_{n-1}\}$ be a set of state-trajectories (demonstrations) such that $\tau_i = (s_0^i, s_1^i, \ldots, s_{T-1}^i), s_t^i \in \mathbb{R}^{d_s}, \tau_i \in \mathbb{R}^{T \times d_s}$. The demonstration set $\mathbb{D}$ results from a set of $n$ human policies $\{\pi_h^i(a|s)|i = 0 : n-1\}$, where the subscript $h$ indicates that it is a human policy, which is inaccessible for us.

We hypothesise that while all human policies maximise the same task reward $R_{task}(s, a)$, each policy is constrained by a preference component. We define the preference component as the state distribution, such that different individuals prefer to visit different states while solving the task. Therefore, human policies are the results of solving the following constrained optimisation problem:

$$\pi_h^i := \arg\max_\pi \mathbb{E}\Big[ \sum_{t=0}^{T-1} \gamma^t R_{task}(s_t, a_t)|\pi \Big], \tag{1}$$

$$\text{subject to } \mathcal{D}_{KL}[p_\pi(s)||g_i(s)] < \epsilon \tag{2}$$

where $g_i(s) = p(s|\pi_h^i)$ is the state distribution preferred by the individual $i$, and $p_\pi(s)$ the state distribution of the learning agent. As different individuals optimise the task reward function with different preferred states, the demonstration distribution can have multiple solution modes.

Our goal is to learn a large set of policies capable of solving the same task in different ways by satisfying different preference components of the demonstrators, rather than learning a single way of solving the task (i.e., mode-seeking policy or mean-seeking policy).Thus, we propose a setup where the learning agent has access to a simulator to receive the task reward from the environment and to observe the true dynamics of the system. At the same time, it receives the demonstrations $\mathbb{D}$, where it needs to optimise its policies for the different unobserved (as it is not explicitly defined) preference components from these demonstrations. In other words, we propose to solve the objective above by combining the optimisation scheme of RL with IL that uses a divergence minimisation scheme.

## 4 SKILL GENERATION FROM BEHAVIOURALLY DIVERSE DEMONSTRATIONS

Our method consists of two steps: (1) latent space modelling from the demonstration, and (2) diverse policy generation on the latent space. In addition, our method allows us to perform *post-hoc solution searching* on this space in order to obtain solutions (policies) that satisfy different constraints.

To find out a large set of policies, one for each $g_i(\cdot)$ as per the KL constraint in Eq. 2, we first need to construct a latent space $\mathcal{Z}$, such that any $g_i(\cdot)$ can be mapped onto that space. Now, if we use a projection function that maps two similar $g_i(\cdot)$s closer to each other in the latent space $\mathcal{Z}$, the KL constraint in Eq. 2 can be approximated using a Euclidean constraint in this space $\mathcal{Z}$. More concretely, for some constant $\delta \in \mathbb{R}^+$, $z_i \in \mathcal{Z}$, and projection function $\text{Enc}(\cdot)$ the problem can be reformulated on the latent space $\mathcal{Z}$ as

$$\pi_i := \arg\max_\pi \mathbb{E}[(R_{task}|\pi)],$$
$$\text{subject to} \quad ||\text{Enc}(\tau_\pi) - z_i||_2 < \delta \tag{3}$$

Practically, we want to find one policy for each $z_i \in \mathcal{Z}$, optimising the objective in Eq. 3, where $z_i$ are uniformly and densely distributed in the space $\mathcal{Z}$. In other words, we want to optimize policies for each *niche* so that the optimized policies maximise the task reward while producing state trajectories specified by its own niche only. This is a *quality-diversity optimization* problem (Cully & Demiris, 2017): finding high-quality solutions according to a cost/fitness function in a space that specifies the *behaviour* of the solutions. We solve this optimisation problem using a quality-diversity algorithm called MAP-Elites (Mouret & Clune, 2015; Vassiliades et al., 2017), where we determine the *behaviour* of the solutions as the latent preferences extracted from our projection function.

### 4.1 LATENT SPACE MODELLING FROM DENSITY ESTIMATION

We first model the latent factors that explain the diversity in the behaviours from the demonstrations. As mentioned in Sec. 3, we hypothesise that the diversity is caused by individual preference over the state visitations; *i.e.*, different individuals prefer visiting different regions in the state space. A similar hypothesis has also been made in previous works such as *max-margin planning* (Ratliff et al., 2006), where the reward function is recovered by matching the state-visitation frequency for every single state and the state-action feature vector. While they use single state visitation counts, we propose to estimate the *state region visitation frequencies* as latent preferences.

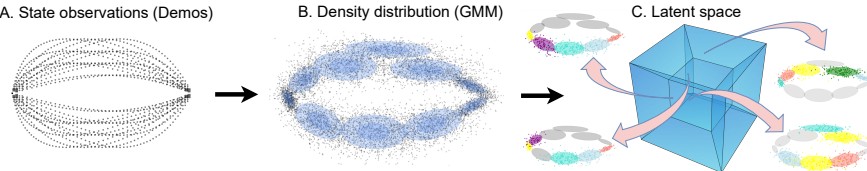

Figure 3: Generation of the latent space from the demonstration. The state observations (A) are modelled by a GMM (B). The latent space (C) is built based on different combinations of mixture components.

Given the state distributions from the demonstrations $p(s)$ as defined in Sec 3, we use a Gaussian Mixture Model (GMM) of $k$ mixture components to model the density as $p(s) = \sum_i^K \phi_i \mathcal{N}(s|\mu_i, \sigma_i)$, where $\phi_i$ is the mixture weight for each mixture component. This state density $p(s)$ models the state observations of all the demonstrations. Under the assumption of a continuous state transition,*i.e.*, states close in time will be in the same state region (see Sec 3), each mixture component will cover a state region in the state space. The state distribution of each individual, $g_i$, belongs to a subset of the entire state distribution. As a mixture component can be seen as a state region, different trajectories would have different state region visitations or assigned mixture components. Given a fitted GMM model, we can estimate the state distribution of each $g_i(s)$ given its observations by a subset of mixture components according to their corresponding posterior probability or *responsibilities* $p(k|s)$. That is, given the sequence of states, we can estimate the responsibilities $p(k|s)$ for these states using the fitted GMM, and use the corresponding mixtures to approximate its state distribution. Given the assigned mixture components $\{k_j\}$, and their assignment frequencies $\frac{f_j}{T}$ (i.e., number of assignment for each component w.r.t the total number of states $T$) for $g_i(s)$, we can approximately define the density of $g_i(s)$ using a new GMM model composed of these mixture components as $g_i(s) \approx \sum \frac{f_j}{T} \mathcal{N}(s|\mu_{k_j}, \sigma_{k_j})$, where the assigned mixture components correspond to state region visitations, and the new mixture weight is readjusted based on their assignment frequencies $\frac{f_i}{T}$.

Now, we can define our projection function, or encoder function $\text{Enc} : \mathbb{R}^{T \times d_s} \to \mathbb{R}^{d_z}, d_z \ll (T \times d_s)$, based on the assignment of the mixture components. Given a set of states from a policy, the GMM is used as an unsupervised clustering method to classify these states into different mixture components. There are different ways to represent this information. For instance, we can have an encoding of size $K$ and give the frequency values of each component $[f_1, f_2, \dots f_k]$. However, to limit the size of the encoding, we choose to take the indices of the $m$ most frequently assigned components without explicit referring to the actual frequencies (detailed algorithm in Appx. Algo 1). The precision of the mapping depends on our choice of $m$ as we will have information loss with a small value of $m$ (further discussion in Appx. A.5.2 and A.4).

Implicitly, we are defining the feature vector of a trajectory as the mixture components that are responsible for its states. The visitation frequency is explicitly taken into account by taking the top $d_z$ most assigned mixture components in descending order. In addition, the use of GMM allows us to evaluate the likelihood of a given trajectory according to the density distribution modelled from demonstrations. For instance, given the example of Fig 4, we can evaluate whether these trajectories belong to in-distribution data or out-of-distribution based on their likelihood (of the state sequence).

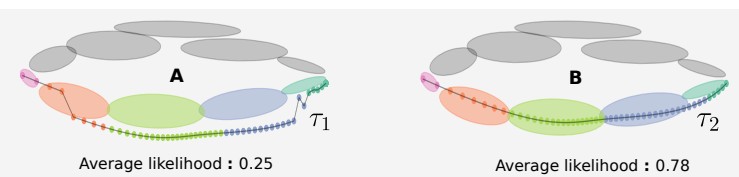

Figure 4: Mixture components/clusters assignment (represented by **coloured** confidence ellipsoids of GMM) for the state distributions of 2 trajectories $\tau_1 \in \mathbb{R}^{80}$ and $\tau_2 \in \mathbb{R}^{50}$. **A.** Out-of-distribution trajectory $\tau_i$ with low average likelihood according to a fitted GMM. **B.** In-distribution trajectory $\tau_2$ with high average likelihood.

**Solution matching**   We can measure the similarity of 2 trajectories by measuring their associated mixture components. As we have the analytical form of each component (*e.g.*, Gaussian with its mean and variance), we can directly measure the distance between their distributions by using measurement of our choice, such as *Bhattacharyya distance* or *KL* divergence.

### 4.2   MAP-ELITES (EA) AS DYNAMICS-AWARE CONDITIONAL GENERATIVE MODEL

Given $\mathcal{Z}$ as the latent space defined by the projection function, we aim to generate a solution archive $\mathcal{P}_z$ parameterised by the latent encoding $z$, where each entry in this archive has a unique encoding $z$. The goal is to generate policies and map them into this space according to encoding values for their respective $g(\cdot)$. While the latent space contains all the possible state distributions defined by the mixture components of the GMM, the solution space is constrained by the system's dynamics, as the possible solutions are subject to the system constraints that cannot generate certain state distributions. Thus, $\mathcal{P}_z \subset \mathcal{Z}$. As the solution archive is the result of filling the latent space with policies, we may use latent space interchangeably to refer to the empty solution space.

EA can be seen as an implicit state density model over states with high fitness/reward (Murphy, 2023). In our case, as EA optimises each instance of encoding, which is associated (deterministically) with a given mixture model, the generation of this space is equivalent to an implicit density modelling conditioned on the latent encodings. Thus, we refer to our method as a conditional generative model that generates policies conditioned on different encodings. The workflow is shown in Fig 5, and the detailed algorithm in Appx. 2.

As mentioned in Sec. 4, we use MAP-Elites as the algorithm of our choice. It starts with sampling $N$ random policy parameters from a uniform distribution: $\theta_{i=0:N-1} \sim \mathcal{U}_{[a,b]}$. Then it evaluates the policies in simulation using the reward function $R_{task}(\cdot, \cdot)$, and at the same time generates the state trajectories $\tau_{i=0:N-1}$, and their corresponding encodings $z_{i=0:N-1}$. Then the policies are inserted into the closest cells in the solution archive $\mathcal{P}$. If two policies compete for the same cell, the one with a higher reward occupies the cell. Once this initialization is done, MAP-Elites randomly selects a policy from the archive, adds a small random noise (mutation) to the parameters, and evaluates the policy on the simulator for the reward, state trajectory, and the corresponding encoding. This new policy is inserted into the archive if either the cell is empty or the new policy has a higher reward; the

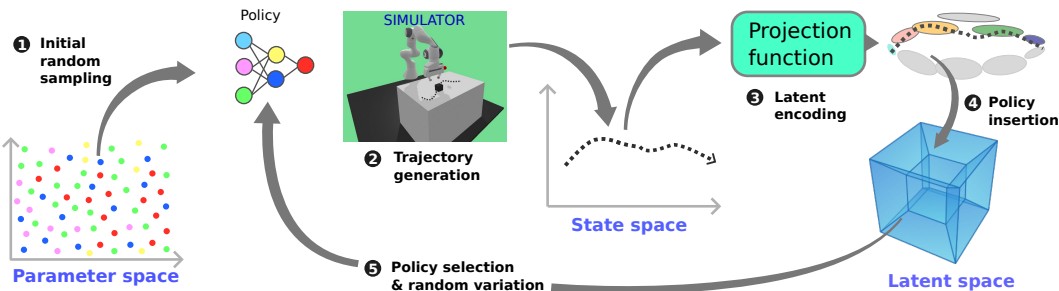

Figure 5: Workflow of policy generation and allocation in the latent space. To initialise, the parameters of the policy (here we use a neural network) are sampled uniformly, and then the policies are executed on the simulator to get the corresponding trajectories (set of states), in order to allocate the policies into the latent space according to their encoding. Then, in a loop, policies are selected from the archive, small variations are added, and new encodings are obtained to insert them into the archive based on their reward.

policy is discarded otherwise. The selection, variation, and insertion continue until the maximum assigned policy evaluation count is reached.

As the highest rewarding policies occupy the nearest cells in the archive, those policies produce maximally rewarding behaviour staying close to the state distribution specified in the cell. In other words, the policies in the archive essentially optimise the objective in Eq. 3 for different $z_i$.

**Post-hoc policy search** Having a set of solutions parameterised by latent encodings allows us to find solutions in a post-hoc manner that satisfy different constraints. For instance, a solution archive $\mathcal{G}$ built for a 2D navigation environment with one obstacle contains solutions for an equivalent environment $\mathcal{J}$ with two obstacles, as the latter space is more restricted, where $\mathcal{J} \subset \mathcal{G}$ (see example in Sec. 5). In addition, given a new demonstration, we can find the closest policy in our archive by finding a policy with the closest distribution distance (see Sec. 4.1 and Alg. 3) according to the encodings.

## 5 EXPERIMENTS

We present 3 experiments with continuous state and action spaces to show that our method is capable of generating a solution archive that optimises implicitly the density distribution from demonstrations. We propose 2 (motion) planning examples where the task is to generate a non-colliding path from two points in an end-to-end manner. The first is a 2D toy path planning environment and the second is a robotic arm motion planning environment using the Franka Emika robot with Pybullet (Coumans & Bai, 2016–2021) as the physic simulator. In both cases, we use only the XY position coordinates as the state observation. For Franka, we fixed the Z axis, as the demonstrations do not contain Z coordinates. Finally, we show a driving experiment modified on (Leurent, 2018) as a close-loop control example, using neural networks for a step-wise decision-making policy. The experiments are run 3 times. The average performance and further ablation study (time, number of valid solutions vs encoding choices) can be found in Appx. A.4. The latent encoding is a vector of 6 dimensions defined by the top 6 most frequent mixture components responsible for the state sequence. A discussion of model selection and reproducibility is included in Appx. A.5.2 and A.3.

**2D path planning** We use 28 non-colliding trajectories with fixed lengths that are generated artificially using *Bézier curves* as shown in Fig 6A. We use *B-spline* as the policy function, which takes 5 control points in total to generate a flexible curve as parameters, where the first and last control points are start and end coordinates, and the resulting points are the parameters of the policy, $\theta \in \mathbb{R}^6$. We use a GMM of 20 components for the multimodal distribution and 10 for the single-mode distributions. Under the exact same setup, we show that the control points are optimised towards state distributions modelled by their respective GMM. As shown in Fig 6, the resulting control points from the single-mode setup are scattered mainly in the region within single-mode state distribution. While for the multi-modal setup, the control points are scattered in both regions of the state space. This comparison is even more noticeable after filtering out the solutions with a higher likelihood (according to their respective GMM models). We show that even though the demonstrations only

cover one solution mode, our method is still capable of discovering multimodal solutions, where the unseen solutions from the demonstrations will have lower likelihood values.

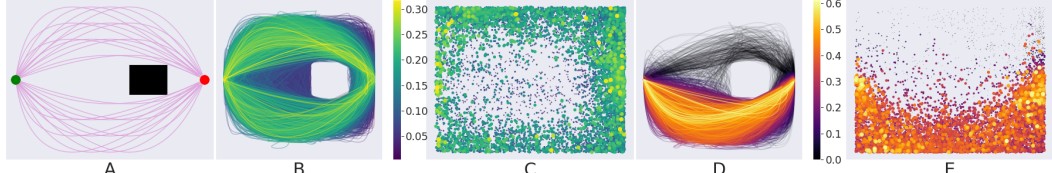

Figure 6: Solution archives optimised for different GMM models fitted to single mode and multimodal demos. **A.** Multimodal demonstration. **B.** Trajectories with GMM from **multimodal demonstration** with colour bar as their corresponding average likelihood. **C.** Control points for policies from B shown with their colour and size re-adjusted according to trajectories's likelihood. **D.** Trajectories from solution space build for the **single-mode demonstrations** (*i.e.*, lower regions) with their average likelihood. **E.** Control points for the policies of Fig. D.

As VAE is commonly used as a feature encoding method as well as a (conditional) generative model, we show its capability of feature encoding and generation ability in this setup. We use the same encoding size, and trained the model for 30000 iterations (details in Sec A.3.1). As shown in Fig 7, for a small dataset, the generator suffers from partial collapse that only generates samples from certain modes and fails to produce diverse samples. The same observation occurs for the encoder, it fails to produce unique encoding values as shown in Fig. 7(B, D). While training the same model using the data generated from our solution, both its capability of reconstruction and encoding improves considerably. However, unlike our projection function, it does not provide an interpretable encoding value. And it cannot take the advantage of the simulator that is available to interact with. Another alternative method is GAN, but it does not work when a small amount of data is provided, and it suffers from the mode collapse even though a large amount of data is available (see Sec 2.1). We show this latter using trajectories from our archive in Appx. A.2).

**Franka arm manipulation** We use the same setup here as in the previous case, except the mixture component to model the state density increases to 25 as more data is available. We collected 140 demonstrations in real life using a motion tracker to track the movement of a pointer to draw the trajectories without considering the real dynamics of the robot. The length of demos spans between 240 and 726 time-steps. This 3D environment contains 3D obstacle, where the resulting trajectories are plotted in 2D space for visualisation purpose in Fig 8.

**Highway driving** We demonstrate here that our method is capable of optimising high-dimensional parameters of neural networks, where $\theta \in \mathbb{R}^{201}$. We collected 40 demonstrations of 151 time-steps using a manual controller from the simulator. Here the observations contain both the position coordinates and the rotation angles of the vehicle, $s \in \mathbb{R}^4$, the action $a \in \mathbb{R}^1$ is the steering applied to the vehicle to overtake the obstacle without going off the road as shown in Fig 9A. We show the resulting behaviours when multimodal and single-mode demonstrations are used in Fig 9.

**Post-hoc solution search** In this case, we show that we can still find valid solutions when the constraints in the environment change. Here, we add another obstacle to the 2D path planning

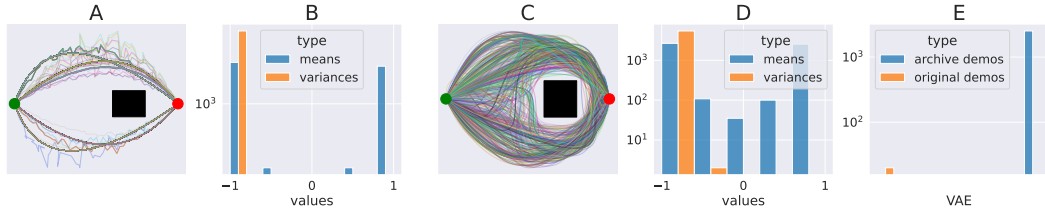

Figure 7: VAE failed on the original demonstrations, but can be fully trained using trajectories from our method. **A.** 1000 samples from VAE trained on the original demonstrations. **B.** Histogram of the predicted encodings for the samples (*i.e.*, mean and variance for the isotropic Gaussian). **C.** 1000 samples from VAE trained on trajectories generated from our solution. **D** Histogram of the predicted encodings for the corresponding demonstrations. **E**. Histogram of the unique encodings from both VAEs.

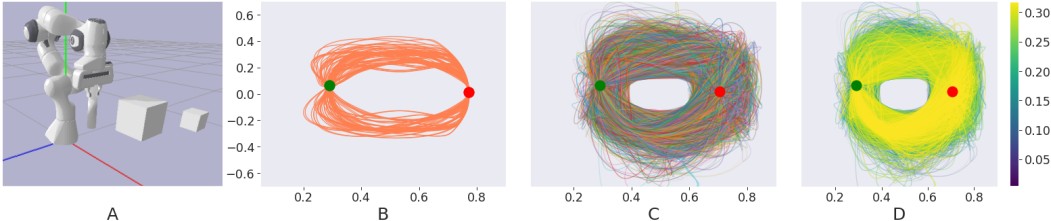

Figure 8: **A.** Franka environment. **B.** Demonstrations of various lengths. **C.** Trajectories from the solution archive. **D.** Trajectories with their corresponding likelihood visualised with a colour map.

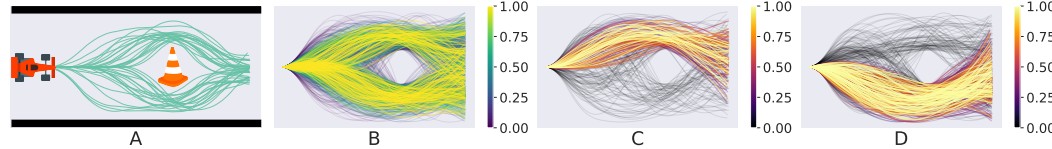

Figure 9: Our method finds feasible driving styles that overtake the obstacle in different ways. **A.** Demonstration. **B.** Trajectories from the multimodal solution archive with the likelihood values. **C/D.** Trajectories from the single-mode solution archives with their respective likelihood values.

environment. By simulating the policies in this modified environment, we obtained 1738 valid trajectories as shown in Fig 10. Notice here that, we can observe more solution modes as results of interpolation between the two existing solution modes from the demonstrations. It is also possible to obtain a set of trajectories close to a given demonstration by matching the latent encodings (see Sec. 4.1) in the solution archive (see Fig 10). Here we compare the result of encoding matching using VAE trained on demonstrations generated by our method (see Fig 7C) and the result from our proposed projection function (more details in Appx. A.3.2). The results show that our projection function enables better matches, as we can measure the distances between the underlying state distributions based on the encoding values (see Sec. 4.1).

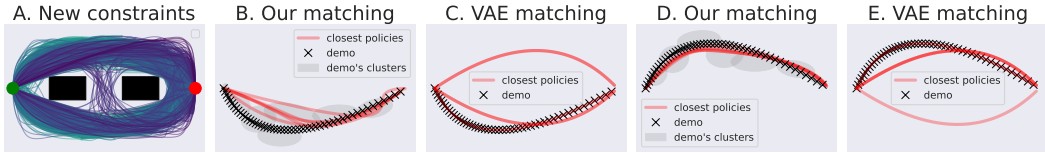

Figure 10: **Post-hoc policy searching in the solution archive to find policies that satisfy different constraint and to find policies for a given demonstration**. **A**: 1738 valid solutions for a new obstacle in the solution archive at Fig 6a. **B/D**: 5 closest policies to a given demo using our proposed projection function with GMM. **C/E**: 5 closest policies using VAE encodings.

## 6   CONCLUSIONS

In this paper, we propose a novel framework that allows us to build a solution space where we generate a diverse range of behaviours from a small set of state-only observations. We show empirically that our method optimises all the policies in the space towards different state density constraints while optimising the task reward. With our proposed projection function that encodes state transitions into meaningful statistical encoding, we can perform a post-hoc policy search in the projected space searching for different policies. Our experimental results show that we are capable of generating multimodal solutions deterministically beyond the provided demonstrations. Further discussion regarding its application use is included in Appex. A.8.

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

## A APPENDIX

### A.1 ALGORITHMS

---

**Algorithm 1** State region visitation frequency

---

1: **function** ENCODE(gmm_model, state_sequence, size_descriptor)
2:     list_components=gmm_model.predict(state_sequences) # Predict the list of mixture components responsible for each state
3:     unique_component, n_counts = **unique**(list_components)
4:     comp_indices = **top_k_components**(unique_component, n_counts, size_descriptor)
5:     **return** comp_indices
6: **end function**

---

**Algorithm 2** Solution archive generation

---

1: **function** ARCHIVE_GENERATE(gmm_model, size_descriptor)
2:     Initialise an instance of policy $\pi$, solution archive $\mathcal{P}$, reward archive $\mathcal{R}$, number of iteration $N$, number of initial random samples $M$
3:     **while** $i \leq M$ **do**
4:         $\theta$ = uniform_sampling(0,1) # sample the parameter vectors from uniform distribution.
5:         $\pi_\theta$ = policy.set_parameter($\pi, \theta$)
6:         $\tau, r_{task}$ = simulate($\pi_\theta$)
7:         $z$ = encode(gmm_model, $\tau$, size_descriptor)
8:         $\mathcal{P}(z) \leftarrow \theta$
9:         $\mathcal{R}(z) \leftarrow r_{task}$
10:     **end while**
11:     **while** $i \leq N$ **do**
12:         $\theta$ = ramdom_solution_select($\mathcal{P}$)
13:         $\theta'$ = mutate($\theta$) # Gaussian noise injection to the parameters.
14:         $\tau, r_{task}$ = simulate($\pi_{\theta'}$)
15:         $z$ = encode(gmm_model, $\tau$, size_descriptor)
16:         **if** $\mathcal{P}(z) = \emptyset$ or $\mathcal{R}(z) < r_{task}$ **then**
17:             $\mathcal{P}(z) \leftarrow \theta'$
18:             $\mathcal{R}(z) \leftarrow r_{task}$
19:         **end if**
20:     **end while**
21:     **return** $\mathcal{P}, \mathcal{R}$
22: **end function**

---

### A.2 KL DIVERGENCE MINIMISATION FOR MULTIMODAL DISTRIBUTION

As KL divergence is not symmetric, the reverse KL and forward KL exhibit very different behaviours. These are also known as *Moment projection* and *Information projection* respectively.

Given the state-action observations from the demonstrations from the expert's policy $\pi_{exp}$, M-projection minimises the following forward KL:

$$\pi_\theta^* = \arg\min_\theta D_{KL}\left(p_{\pi_{exp}}(s,a) \,||\, p_{\pi_\theta}(s,a)\right),$$

$$D_{KL} = \int p_{\pi_{exp}}(s,a) \log\left(\frac{p_{\pi_{exp}}(s,a)}{p_{\pi_\theta}(s,a)}\right) d(s,a) = \mathbb{E}_{p_{\pi_{exp}}(s,a)}\left[\log\left(\frac{p_{\pi_{exp}}(s,a)}{p_{\pi_\theta}(s,a)}\right)\right]. \quad (4)$$

While the I-projection minimises the following reverse KL:

$$\pi_\theta^* = \arg\min_\theta D_{RKL}\left(p_{\pi_{exp}}(s,a) \,\|\, p_{\pi_\theta}(s,a)\right),$$

$$D_{RKL} = \int p_{\pi_\theta}(s,a) \log\left(\frac{p_{\pi_{exp}}(s,a)}{p_{\pi_\theta}(s,a)}\right) d(s,a) = \mathbb{E}_{p_{\pi_\theta}(s,a)}\left[\log\left(\frac{p_{\pi_{exp}}(s,a)}{p_{\pi_\theta}(s,a)}\right)\right]. \tag{5}$$

For M-projection, the difference between $p_{\pi_{exp}}(s,a)$ and $p_{\pi_\theta}(s,a)$ is weighted by $p_{\pi_{exp}}(s,a)$. Which means that when $p_{\pi_{exp}}(s,a) = 0$, the discrepancy of $p_{\pi_\theta}(s,a) > 0$ from $p_{\pi_\theta}(s,a)$ will be ignored. Mathematically, this means that $p_{\pi_{exp}}(s,a)$ will typically over-estimate the support of $p_{\pi_\theta}(s,a)$, due to $p_{\pi_{exp}}(s,a) > 0$ whenever $p_{\pi_\theta} > 0$ to make sure KL divergence stays finite. As result, the projecting exhibits a mean-seeking, also known as moment-matching, behaviour that averages over several modes given a multimodal distribution. While it avoids the low probability assignment of the region with data (*e.g.*, the other mode of the distribution), it inevitably assigns probability mass to non-data region (Theis et al., 2015) as shown in Fig 2.

For I-projection, as difference between $p_{\pi_{exp}}(s,a)$ and $p_{\pi_\theta}(s,a)$ is weighted by $p_{\pi_\theta}(s,a)$. $p_{\pi_h}(s,a)$ will typically under-estimate the support of $p_{\pi_\theta}$ and concentrate on one of its modes that results in a mode-seeking behaviour. This is due to $p_{\pi_{exp}}(s,a) = 0$ whenever $p_{\pi_\theta}(s,a) = 0$ to make sure KL divergence stays finite. Using this projection is not straightforward as we might only have access to samples from the distribution rather than the data density itself (Murphy, 2023), where M-projection can be done easily by maximising the average log-likelihood with respect to the given training dataset as an equivalent way of minimising the KL.

### A.2.1 EXPERIMENTAL ANALYSIS

Given the multi-mode and single-mode demonstrations of the 2D path planning environment, we proposed in Sec. 5. We show the results of Gaussian Mixture Regression (GMR) model, build on the same GMMs we used for our method (see A.4). While it performs well in the single-mode demonstrations, the result from the multimodal distribution exhibits a mean-seeking behaviour as shown in Fig 11.

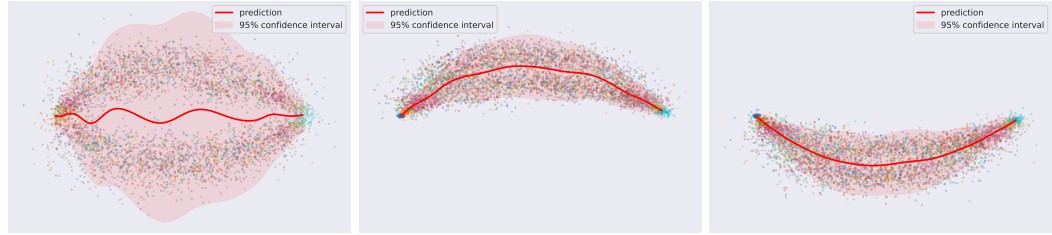

Figure 11: Results of GMR on multimodal demonstrations and single-mode demonstrations. **Left**: Prediction with 95% confidence interval from a GMR trained on multimodal demonstrations. **Middle/Left**: GMR predictions on single-mode demonstrations.

Extending the discussion in Sec 2.1 about GAN in dealing with multimodal distribution. We show experimentally the performance of GAN in generating trajectories for the 2D path planning environment. As GAN cannot deal with a small dataset, we trained it using the trajectories generated by our archive, which contains 3881 trajectories. The results are shown in Fig 12. The discriminator takes the entire trajectory as the input and has 3 hidden layers with 128, 32, and 8 neurons respectively. The generator takes an input of a 7-dimensional vector and uses 3 hidden layers with 128, 64, and 32 neurons respectively to generate the trajectory of the same size. The model is trained for 23000 iterations. The results are consistent with mode-seeking behaviour where it converges to one of the solution modes discussed in Sec 2.1.

### A.3 EXPERIMENT SETUPS AND REPRODUCIBILITY

We train all the GMM models using *sklearn* library with a fixed random state to ensure reproducibility. The experiments were run 3 times per setup (*e.g.*, multimode and single-mode demonstrations), with 2 different types of encodings. We show the average result for single-mode demonstrations and

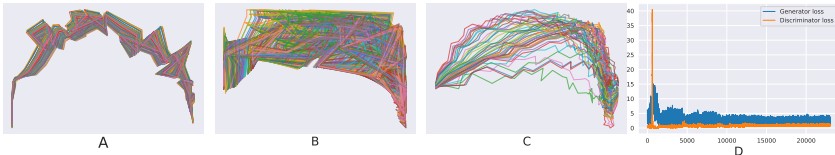

Figure 12: **A**: Samples from GAN after 3000 iterations. **B.** Samples from GAN after 15000 iterations. **C.** Samples from GAN after finished training. **D.** Training loss of the discriminator and generator during training.

| Experiments | iterations | Real time | CPU time |
|---|---|---|---|
| Planning (multimodal) | $10^5$ | 3m49s $\pm$ 1m6s | 13m1s $\pm$ 0m28s |
| Planning (upper) | $10^5$ | 2m33s $\pm$ 0m12s | 11m51s $\pm$ 1m18s |
| Planning (lower) | $10^5$ | 2m27s $\pm$ 0m44s | 12m3s $\pm$ 0m21s |
| Franka | $10^5$ | 6h49m $\pm$ 11m57s | 6h49m $\pm$ 11m57s |
| Highway (multimodal) | $5 \times 10^5.$ | 2h46m $\pm$ 4m19s | 13h29m $\pm$ 36m14s |
| Highway (upper) | $5 \times 10^5.$ | 1h54m $\pm$ 0m56s | 12h35m $\pm$ 4m14s |
| Highway (lower) | $5 \times 10^5.$ | 1h26m $\pm$ 11m57s | 12h36m $\pm$ 42m9s |

Table 1: Training time

multimodal demonstrations. As most of the experiments allow us to run parallel threads. We record both the average true elapsed time and the average CPU time, with their standard deviation. The results are shown in Table 1. For 2D path planning, we use Intel Core i7-6850K CPU @3.60GHz with 6 cores and trained our model using 12 parallel threads. For Franka, we use a single thread process with 1 CPU core of Xeon Gold 6148 @2.40GHz, and 16GB of RAM without multi-threading due to the high memory demand from the physical simulator. For the highway environment, we use 8 cores of Intel Xeon Gold 6248 @2.50GHz with 12 parallel threads and 8GB of RAM.

In the highway environment, we use neural networks with 2 hidden layers of 8 and 16 neurons respectively as policy. The total number of parameters is 201 elements. The parameters of the networks are purely optimised using random sampling and perturbation from EA without any back-propagation. As we use the exact setup from 2D path planning in the Franka environment to show the applicability of our methods in "real" applications besides the 2D path planning experiment, we only test the result in the multi-modal setup. To ensure maximum reproducibility, the solution archive is initialised with $10^4$ solutions with uniformly sampled parameters from a given parameter space to minimise the advantage of a good initialisation. The parameter space for 2D path planning is defined within the range of $[-5, 5]$, while for Franka it is defined within $[0, 1]$ and for highway within $[-1, 1]$.

As we tested 2 different types of encoding, we report the performance results in the ablation study (see Appx. A.5), where we discuss the impact of the size of the encodings vs the size of the solution archive, among the others.

### A.3.1 SETUP FOR VAE

We use a VAE with 5 hidden layers with 256,128,64, 64,32 neurons correspondingly. Unlike our projection function, the VAE cannot handle data of different sizes, instead, it receives the full trajectory with a fixed length. Thus, it cannot deal with the demonstrations intended for Franka Arm environment, as they have various sizes.

### A.3.2 POLICY SEARCH USING ENCODING MATCHING

Given a demonstration, we search for the closest policy by finding the one with the minimal distance in the encoding space. The algorithm can be found in Alg. 3.

To compare the result of VAE with our method, we use the samples generated by VAE (see Fig. 7C),

---

**Algorithm 3** Policy searching

---

1: **function** POLICY_SEARCH(archive_policies, archive_encodings, encoding_demo, n_matches)
2:     Initialise an array of distances $\mathcal{D}$
3:     **for** encoding in archive_encodings **do**
4:         $d = \text{distance\_function(encoding, encoding\_demo)}$
5:         $\mathcal{D} \leftarrow d$
6:     **end for**
7:     closest_policies = **top_n_closest_policies**(archive_policies, $\mathcal{D}$, n_matches)
8:     **return** closest_policies
9: **end function**

---

### A.4 MODEL SECTION OF GMM

The model selection, *i.e.*, the number of GMM components used to model the state density distribution from the demonstration, is determined by Bayesian Information-theoretic criteria (BIC). We choose the model with lower BIC as the common practice. The GMM used for the 2D path planning environment is shown in Fig 13 using confidence ellipsoid. The GMM used for the Franka environment is shown in Fig 14. We omit the visualisation for the highway environment, as we use 4D Gaussian which cannot be visualised with confidence ellipsoid.

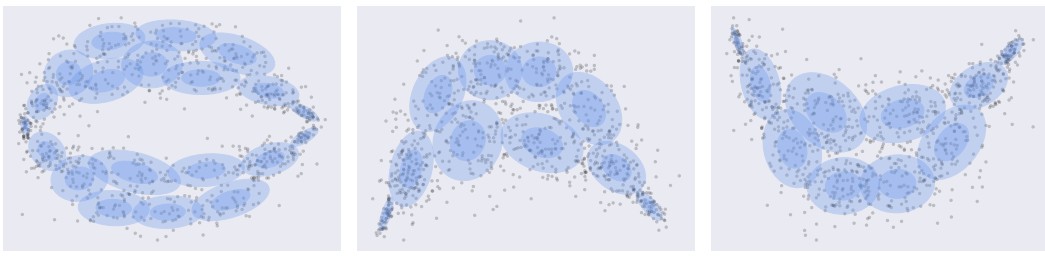

Figure 13: GMM models used for 2D path planning environment. **Left**: GMM confidence ellipsoid for the multimodal demonstrations. **Middle**: GMM confidence ellipsoid for the single-mode demonstrations (upper region). **Right**: GMM confidence ellipsoid for the single demonstrations (lower region).

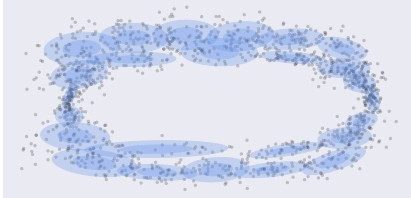

Figure 14: GMM model for the multimodal demonstrations recorded using motion tracker.

### A.5 ABLATION STUDIES AND NUMERICAL QUANTIFICATION OF SOLUTION ARCHIVE

In this section, we analyse the performance of our method by analysing the resulting solution archive generated by our method. We include as the performance indicator the total number of policies generated and the total valid solutions that reach the given task reward, *i.e.*, reaching the goal while preventing the collisions. We report the results under different setups.

### A.5.1 COMPARISON OF DIFFERENT ENCODINGS

We compare 2 different type of encoding: 1) Encoding with the top 6 most frequently assigned mixture component as discussed in Sec 4.1, shown in Table 2. 2) Same encoding but with an

additional likelihood value added. This latter is proposed to avoid the case of having two different trajectories with the same encoding but with different likelihood values (i.e., out-of-distribution trajectories that happened to be mapped to the same encoding), shown in Table 3. In the case of the point environment, the inclusion of likelihood does not increase the size of the final solution archive. We hypothesise that this is caused by the segmentation of the solution space, as the average likelihood is low in general for all the trajectories (see the likelihood values for the trajectories in Fig 6), it worsens the uniform segmentation of the space. In reality, while different trajectories can have the assigned mixture components, their respective frequencies may rarely coincide. For instance, in Fig 4, two trajectories can have the same assigned component with low and high average likelihood respectively, but their resulting encodings are different due to the different frequency values. Thus, these trajectories will not compete for the same *niche* in the solution space. While in the case of highway environment, the inclusion of the likelihood does increase the size of the descriptor as the average likelihood values are higher, which span within [0, 1] (see the colour map for the trajectories in Fig 9), which allows a uniform segmentation of the archive space.

| Experiments | Demos's size | Encoding/GMM[*] | Total solutions | Valid |
|---|---|---|---|---|
| Planning (multimodal) | $28 \times 50$ | 6/20 | $7703 \pm 20$ | $6584 \pm 38$ |
| Planning (upper) | $14 \times 50$ | 6/10 | $2329 \pm 47$ | $1999 \pm 27$ |
| Planning (lower) | $14 \times 50$ | 6/10 | $3920 \pm 29$ | $3314 \pm 34$ |
| Highway (multimodal) | $40 \times 151$ | 6/20 | $3900 \pm 59$ | $1009 \pm 42$ |
| Highway (upper) | $20 \times 151$ | 6/10 | $1657 \pm 33$ | $464 \pm 16$ |
| Highway (lower) | $20 \times 151$ | 6/10 | $1187 \pm 14$ | $261 \pm 9$ |

Table 2: Number of demonstrations vs archive solutions with encoding without likelihood. [*]Number components for the encoding / total number of mixture components.

| Experiments | Demos's size | Encoding/GMM[+*] | Total solutions | Valid |
|---|---|---|---|---|
| Planning (multimodal) | $28 \times 50$ | 7/20 | $3810 \pm 16$ | $3554 \pm 6$ |
| Planning (upper) | $14 \times 50$ | 7/10 | $2685 \pm 12$ | $2415 \pm 15$ |
| Planning (lower) | $14 \times 50$ | 7/10 | $3761 \pm 27$ | $3270 \pm 7$ |
| Franka | $140 \times$ (various) | 7/25 | $8581 \pm 24$ | $6948 \pm 48$ |
| Highway (multimodal) | $40 \times 151$ | 7/20 | $4781 \pm 23$ | $1116 \pm 25$ |
| Highway (upper) | $20 \times 151$ | 7/10 | $1919 \pm 64$ | $515 \pm 25$ |
| Highway (lower) | $20 \times 151$ | 7/10 | $1237 \pm 40$ | $260 \pm 11$ |

Table 3: Number of demonstrations vs archive solutions with encoding that has likelihood as an additional dimension. [+]Here we have 6 mixture components plus one value for the average likelihood.

Approaches as Tanwani et al. (2021) has used Hidden semi-Markov Models to extract the temporal sequence of the demonstration for imitation learning, which takes the transition probability between different mixture models into account. Our encoding does not consider the transition probability between different mixture components; the use of Hidden Markov Models (HMM) could be done as a potential future work.

### A.5.2 COMPARISON OF DIFFERENT CHOICES FOR GMM

**Encoding space dimensionality.** Given a GMM with $d_{gmm}$ components and an encoding size of $d_z$. The number of possible encodings is defined by different permutation of $d_{gmm}$, its size has a theoretical upper bound of $\frac{d_{gmm}!}{(d_{gmm}-d_z)!}$. In practice, its real upper bound depends on the system's dynamics, *e.g.*, it may be impossible to have a state distribution with only 2 different state regions that are far away from each other as the state transition is continuous.

As discussed, the size of the latent (encoding) space depends on the number of mixture components that represent different state region visitation and the size of the descriptor to choose. Here we show the result of using 2 different GMMs with 10 and 5 mixture components respectively that have higher BIC scores compared to the GMM with 20 components (see model selection in A.4).

For the GMM with 10 components, we keep the size of descriptor unchanged, *i.e.*, 7, while for the GMM of 5 components, we reduced it to 4 as the total number of components is reduced. For the former case, the number of total solutions is reduced to 1163 trajectories with 854 valid trajectories on average for the former case. While for the latter case, the sizes reduces to 120 total trajectories and 105 valid trajectories respectively. This is due to the fact that a lower number of mixture components will reduce the size of the latent space. The result is shown in Fig 15.

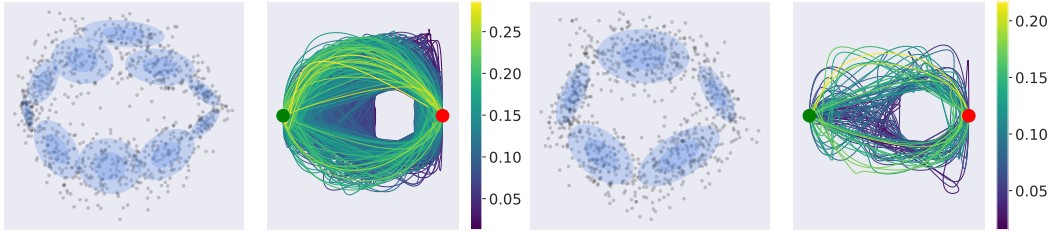

Figure 15: Comparison of solution archives optimised with different size of GMM mixture components in the projection function. **Left**: Mixture component with 10 components, and the resulting solution archive with valid trajectories shown with colour map that corresponds to their respective likelihood values. **Right**: Mixture component with 5 components and the associated solution archive with valid trajectories with their respective likelihoods.

### A.6    POST-HOC SOLUTION SEARCH FOR CHANGING SIMULATOR

In this section, we show that the solution archive trained on the toy 2D path planning environment with 2D obstacles contains valid solutions for the Franka arm environment with 3D obstacles. Where the former does not have any physic engine or accurate collision detection mechanism; while the latter is a high-fidelity simulator with a real physic simulator and sophisticated collision detection. And the state space for the 2D environment is defined to be the same as the Franka environment.

We configure the toy environment to have the approximate location of the obstacle. In the case of the Franka environment, the obstacle is a 3-dimensional cube instead of a 2D obstacle (see Fig 8A). Thus, the collision detection of the toy environment is far from perfect compared to the Franka environment. We use the same parameters set up as detailed in Sec 5 and Appx. A.3. The GMM is fitted to the demonstrations recorded for the Franka arm (see Fig 14). Training on a simplified simulator has its advantage such as faster convergence speed; it reduces the training time as the simulation time is faster in the toy environment. As we can see from Table 1, the training time for Franka ($6h49m$) is considerably longer than the 2D environment ($3m49s$) due to the slow physic simulation. By allowing access to the simulator, we can filter out solutions by simulating the policies from the trained solution archive in the environment to validate them under different constraints. In this case, the constraints are different due to the change in the environment. The final solution archive trained in the 2D toy environment contains 9076 solutions, with 5978 valid collision-free solutions for the Franka environment that reach the goal location.

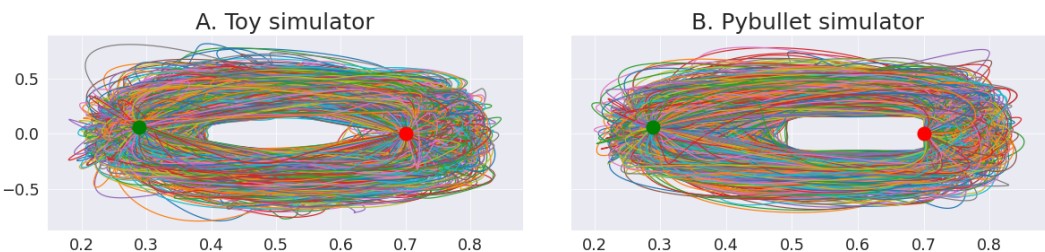

Figure 16: Solution archive trained using toy environment deployed in the 3D Franka Arm environment with Pybullet engine. The resulting trajectories are plotted in 2D space for better visualisation. **A**: Valid solutions for the toy environment with 2D obstacles without physic engine. **B**: Valid solutions for Franka Arm environment with Pybullet as physic engine with 3D obstacle collision detection.

### A.7 Additional multimodal experiment

In addition to bimodal demonstrations for the 2D path planning in Sec. 4.1, we include a new scenario where more solution modes are present in the demonstrations to show that our method can deal with more solution modes. Here we use a total of 12 demonstrations generated using sine and cosine functions and used the same parameters' setup as the previous experiment. The GMM takes 20 mixture components, and we use a 6-dimensional encoding value. The resulting solution archive contains 6309 trajectories as shown in Fig 17. While the solution archive built from bimodal demonstrations contains also multiple solution modes such as interpolation results, these demonstrations present with lower likelihood values as shown by its colour map (see Fig 6, 10A). However, in this four-mode solution case, we can observe clearly that the intersecting trajectories have higher likelihood values due to the presence of data points in the demonstrations.

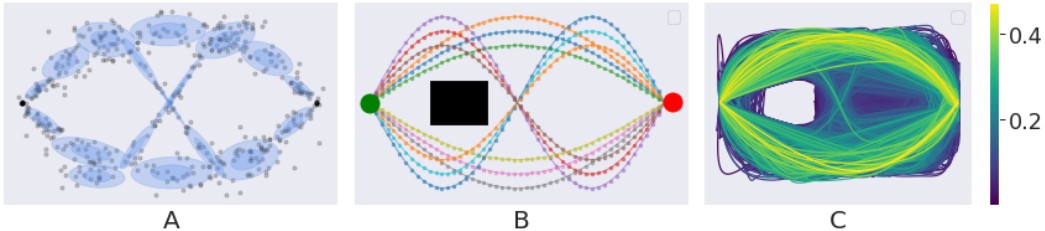

Figure 17: **A**: GMM fitted to demonstration **B.** 12 multimodal demonstrations with 4 solution modes. **C.** Valid solution archive with trajectories colour with their respective trajectory likelihood.

### A.8 Potential applications

We dedicate this section to discussing the applicability of our method. One of the potential applications is data augmentation. Given a small set of state-only demonstrations, our method is capable to generate diverse state-action demonstrations. These can be used, for instance, to provide more data input to other approaches. One example is shown in the performance analysis of VAE in Fig 7, VAE suffers from mode collapse when it is trained using a small dataset. However, our method can generate a large set of demonstrations to train the VAE to produce more diverse samples. However, notice here that in this case, VAE does not necessarily extrapolate beyond the provided demonstrations.

In practice, a more useful use case is the recollection of visual demonstrations in a high-fidelity environment. For instance, as shown from our Franka Arm environment, we are capable to generate a large set of demonstrations. While our approach currently does not support image-based input, it can be used to generate training data for image-based methods by recording the resulting trajectories in the simulator.

Another potential application is the search for policies that satisfy different constrains, such as constraints from different environments or policies that are capable of producing certain behaviours using encoding matching, as shown in Appx. A.6 and Sec. 5.

