# OpenReview forum: "Dynamics-aware Skill Generation from Behaviourally Diverse Demonstrations"
_ICLR.cc/2023/Conference — Submitted to ICLR 2023_

### Official Review · Reviewer_KHmd · 2022-10-25

**Confidence:** 3
**Correctness:** 3
**Technical Novelty And Significance:** 3
**Empirical Novelty And Significance:** 3
**Recommendation:** 6

**Clarity, Quality, Novelty And Reproducibility:**

The overall clarity, quality, and novelty of the paper are good. The paper also gives experiment details to reproduce the results.

**Strength And Weaknesses:**

Strength:
The approach or the combination of approaches is novel and the experiments clearly show its capability to learn from diverse demonstrations.

Weaknesses:
Would be better to have more ablation studies on the algorithm design choices, e.g. compare some different projection methods, as well as different generative models.

Would make the results more convincing by comparing with more than one baseline, especially other approaches that are designed for multimodal behaviors (e.g. VAE with multimodal latent space distributions).

**Summary Of The Paper:**

This paper proposes a novel algorithm to learn diverse behavior styles from demonstrations. It uses a projection function to encode demonstration trajectories into latent spaces and then uses MAP-ELITES to find a set of possible policies. Experiment results how the proposed model is able to learn a better coverage of different behaviors.

**Summary Of The Review:**

The paper proposes a novel framework and the experiment results clearly show its capability to learn diverse behavior styles.

---

> ### Author Response · Authors · 2022-11-19
> **Authors' response**
>
>
> ***“Would be better to have more ablation studies on the algorithm design choices, e.g. compare some different projection methods, as well as different generative models.” “Would make the results more convincing by comparing with more than one baseline, especially other approaches that are designed for multimodal behaviors (e.g. VAE with multimodal latent space distributions).”***
>
> - The ablation studies (different choice of encoding, GMM model selections, etc, in affecting the quality of the solution archive) can be found in Appendix 6. However, it was under a different name which might cause the misunderstanding. We have changed its title now to ”ABLATION STUDIES AND NUMERICAL QUANTIFICATION OF SOLUTION ARCHIVE“  to make this information more accessible.
>
>
> - Generative models would fail in our setup due to lack of data, as we are dealing with a very small dataset.  We agree that sophisticated methods would be able to recover multimodal distribution if a large scale of data is available. This is consistent with our experiment shown in our analysis of vanilla VAE in Fig. 7. Without having access to a large set of demonstrations, it suffers from mode collapse.  While using behaviours generated by our method, it is capable to recover more behaviours (Fig. 7C). However, the generation of multimodal behaviours is a result of sampling from a learnt Gaussian distribution. Therefore, the behaviours cannot be generated deterministically like our approach, either be possible to search policies that satisfy different constraints as we propose. We also included a standard GAN (Fig.12) in appendix 2 as an alternative generative model. As GAN optimises the JS divergence, it results from mode-seeking behaviours that converge to one single solution mode. Additional mechanisms would be needed, and a larger dataset is required in addition to stabilising the convergence of the generator and the discriminator. The theoretical analysis regarding why optimising different divergences would not lead to multimodal distribution is provided in Sec 2.1 and Appendix A.2), the same as the discussion about GAN-based networks in dealing with the setup we proposed here.
> In addition, generative models would only be applicable for path planning examples as they cannot be served as policy $\pi(a\vert s)$ that maps states to actions. Theoretical discussion about why current LfD methods would fail in the setup we proposed is also included in Sec.2 and Appx 2. In addition, existing LfD methods do not allow us to perform post-hoc solution search or policy matching from demonstration, as we proposed in our method.

---

### Official Review · Reviewer_9jjm · 2022-10-25

**Confidence:** 3
**Correctness:** 3
**Technical Novelty And Significance:** 2
**Empirical Novelty And Significance:** 2
**Recommendation:** 3

**Clarity, Quality, Novelty And Reproducibility:**

* The paper is well organized but some parts of the proposed approach is not easy to understand. As the proposed method consists of multiple training stages and components, (1) an illustration of the entire algorithm and (2) a better overview of the proposed method could be very helpful to improve the readability.

* The experiments well describe how the proposed method can generate a set of diverse behaviors. However, the experiments are done in too simple environments (2D or 4D observation and action spaces) and the demonstrations include only two modes.

* This paper tackles an important problem of learning diverse behaviors from data and the proposed method looks novel.

**Strength And Weaknesses:**

### Strengths

* This paper tackles an important problem of learning policies that perform multi-modal behaviors.

* The experiments demonstrate that the proposed method can acquire policies that correspond to diverse behaviors.


### Weaknesses

* The proposed method uses a Gaussian Mixture Model to approximate the state distribution. This may work on a small state space but would not scale to a high-dimensional state space, which is more practical. Moreover, this does not entail temporal information of the trajectory, which can limit its applicability to complex behaviors (e.g. loop).

* There are some alternative approaches to achieve multi-modal behaviors without need for the simulator and task reward [a,b]. Discussion about comparisons to [a,b] would be great.

* The tested environments and tasks are very simple -- 2D path planning with bimodal trajectories.

* The experiments tested on datasets with only two modes. It would be great to verify the proposed method with more modes.

* Instead of comparing with vanilla VAE, a comparison to VQ-VAE or VAE with a discrete latent space could be more appropriate since a quantized embedding in the proposed method can be the main source of generating multi-modal trajectories.

* The paper claims that the proposed projection function provides the interpretable encoding value. However, the encodings shown in the paper are interpretable mainly because the experiments are done in 2D path planning. The proposed encoding will not be interpretable with the high-dimensional observations.

* The proposed method assumes access to a simulator and a task reward, which limits its scalability to many applications.




[a] Shafiullah et al. Behavior Transformer: Cloning k modes with one stone. NeurIPS 2022

[b] Wang et al. Diffusion Policies as an Expressive Policy Class for Offline Reinforcement Learning. arXiv 2022

**Summary Of The Paper:**

This paper proposes a method that learns a set of policies that can cover all modalities of the demonstration dataset. The proposed method first models the state distribution of the demonstrations using a Gaussian Mixture Model (GMM). In this paper, a trajectory is represented with indices of the top $d_z$ mixture components that best describe state distributions of the trajectory. Then, this paper searches for policies that best represent each of the trajectory representations using an evolutionary algorithm, which randomly samples policies, rollouts the policies, encodes resulting trajectories into trajectory representations, stores the best policy for each trajectory representation, and iterates this process. Since each trajectory representation holds a policy that generates the corresponding trajectory, this method can achieve a set of policies that generate multi-modal trajectories. The experiments show that the proposed method can generate bimodal trajectories.

**Summary Of The Review:**

This paper tackles an important problem of learning multi-modal behaviors from demonstrations. However, the proposed approach does not seem to scale and the evaluation is done on too simple tasks to show its impact and practical use. Thus, I am recommending rejecting this paper.

---

> ### Author Response · Authors · 2022-11-19
> **Authors' response (2/2)**
>
>
> ***“The tested environments and tasks are very simple -- 2D path planning with bimodal trajectories” “The experiments tested on datasets with only two modes”***
>
> - We did include more experiments besides the 2D path planning environment. We tested our method also in a 3D Franka Arm environment with Pybullet as the physic engine and a 2D driving environment. Where the former two receive only 2D observations (x-y position coordinates) as data input, and the latter receives 4D observations (position coordinates and rotation angles), We've updated the experimental section for better clarification.  We agree that our results are visualised on 2D observation space and overall use low-dimensional data. The reasoning behind this choice is that our experiments focus on showing the feasibility of multimodal policy generation that go beyond the provided demonstrations - discovering solution modes such as interpolation between the demonstrated ones. It is easier to visually multi-modal trajectories in 2D space than in higher dimensional observation space. Nevertheless, Franka Arm with Pybullet simulator is a very widely used environment for the RL community.
>
>
> - In the new update, We have included experiments with more than two and related discussion in Appendix 7, where we show that we are capable of dealing with more than bimodal trajectories. However, notice that the solution archive built from bimodal demonstrations contains more than 2 solution modes. This might not be obvious due to the overlapping trajectories in the plot.  We’ve updated figure 10 to better highlight the trajectories that result from the interpolation of existing bimodal demonstrations.
>
>
>
> ***“The paper claims that the proposed projection function provides the interpretable encoding value. However, the encodings shown in the paper are interpretable mainly because the experiments are done in 2D path planning. The proposed encoding will not be interpretable with the high-dimensional observations.”***
>
> - Apologise for misunderstanding the term "interpretability" in our context. Our notion of interpretability is that by looking into the encoding vector, we can infer how often a specific state-space region is visited by a trajectory compared to the other state-space regions. Our notion of interpretation has no relation to whether we can plot a trajectory on a 2D surface or not. We claim that the projection function provides interpretable results, as we can measure the similarity of behaviours using the encoding values (that represent directly different Mixture components, i.e., Gaussian distribution, of the GMM), thus, performing the post-hoc policy search. The visualisation for 2D path planning is just for understanding purposes to show that indeed the latent encoding can be interpreted in the state distribution space. The encoding will be interpretable in the sense that we know each element correspond to a Gaussian distribution, thus we have direct access to the overall state distributions. This result is interpretable compared to the encoding generated by some other methods, i.e., VAE, where we cannot “interpret” the result.
>
> - Moreover, under this encoding mechanism, we can choose an encoding value without having an actual demonstration. Let's say we have a GMM fitted to the given expert demonstrations. If we would like to have a policy that produces states from $k$ state regions, we can easily construct an encoding vector without having a demonstration as the reference.  Searching for the policy in the solution archive is then done by matching the encoding values.

---

> ### Author Response · Authors · 2022-11-19
> **Authors' response (1/2)**
>
>
> ***[Summary of the paper] The experiments show that the proposed method can generate bimodal trajectories.***
>  experiments show that the proposed method can generate bimodal trajectories.
>
> The generated policies can produce more than just bimodal solutions. The solution archive also contains interpolation results and further diverse behaviours. In fact, the resulting solutions expand all the possible states, leading to richer behaviours than the original demonstrations. Due to the high density of the solutions and the overlapping states, it can be difficult to visualise it in our figures. For better visualisation, we have updated, for instance, Fig. 10 to better show that the solutions actually contain interpolation results. Furthermore, as numerical quantification, table 2 and table 3 in the appendix show the size of the resulting solution archive and the original demonstration size. For instance, for the 2D path planning, we obtain on average 3810 trajectories using only 28 demonstrations as input.
>
>
> ***“There are some alternative approaches to achieve multi-modal behaviors without need for the simulator and task reward [a,b]. Discussion about them would be great.” “The proposed method assumes access to a simulator and a task reward, which limits its scalability to many applications.”***
>
>
>
> - Neither [a] nor [b] would be applicable in our setup: multimodal demonstration with state-only observations that only contain very few demonstrations. We’ve updated the paper discussing these 2 in Sec. 2.2 as requested. Here is a detailed discussion: [a] uses diffusion models that can only be used for path planning, while our method can be used for control problems as shown in the driving experiment. While [b] addresses similarly the multimodal demonstrations problem, it only deals with state-action observations rather than state-only observations proposed in our paper. Thus, it was not originally discussed in this paper. Moreover, neither of these can extrapolate beyond the provided demonstrations either allow us to perform the post-hoc policy search, while our approach can generate behaviours that go beyond the provided data and allow us to find policies that satisfy different constraints.
>
> - The use of the simulator is required for the setting of Imitation Learning from Observations alone (ILFO). Under this setup, supervised or unsupervised learning approaches cannot be applied directly to find a policy, and the agent must interact with the environment or with a simulator [1]. Moreover, access to a simulator is also assumed in the current SOTA for LfD, which are essentially Adversarial-based methods (i.e., SAIL, GAIL, etc). Study [2] suggested that BC without access to a simulator has lower performance than the adversarial-based methods. Discussion about these approaches can be found in the same section and Appx. 2. Notice that, adversarial methods suffer from other problems such as the difficulty of stabilising the convergence of both generator and discriminator, among others (Discussion also included Sec. 2 and Appx. 2).
>
> - The requirement for the reward function is justified in our problem setup. We are proposing a new Imitation-guided RL framework that formulates the problem of multimodal policy generation as a constrained optimisation problem (See Sec.3). Current IL methods try to answer the question of *how to mimic the existing (multimodal) demonstrations*. However, we try to answer the question of *how can we discover more behaviours from existing (multimodal) demonstrations given that the task reward function is known*. Furthermore, we do not assume a large set of demonstrations which other approaches assume, that is essentially why adversarial-based methods would not solve the problem of learning from multimodal demonstration we proposed here. As it needs a large set of data in order to stabilise the convergence between the discriminator and generator and prevent the problem of mode collapse as discussed in Sec. 2.1.
>
> - Moreover, existing approaches do not allow us to perform the post-hoc policy search that we proposed in our paper.
>
> - We have updated the paper to clarify and justify all these points mentioned above.
>
>
> [1] Sun, Wen, et al. "Provably efficient imitation learning from observation alone." International conference on machine learning. PMLR, 2019.
>
> [2] Ghasemipour, Seyed Kamyar Seyed, Richard Zemel, and Shixiang Gu. "A divergence minimization perspective on imitation learning methods." Conference on Robot Learning . PMLR, 2020.

---

### Official Review · Reviewer_NygL · 2022-10-25

**Confidence:** 4
**Correctness:** 3
**Technical Novelty And Significance:** 2
**Empirical Novelty And Significance:** 3
**Recommendation:** 6

**Clarity, Quality, Novelty And Reproducibility:**

This description of work is relatively clear. The technological and practical novelty are not good enough (mainly due to these are not detailed well by the authors). Reproducibility is unknown because the codes are not open.

**Strength And Weaknesses:**

Strength: writing and novelty
Weakness: Reproducibility and application potential


**Summary Of The Paper:**

This study built a parameterized solution space to help model driving behaviors for planning, which considering the effect of individual differences of driving demonstrations.

**Summary Of The Review:**

1. How about planning efficiency? Planning time consumption is vital to driving safety, and generally, an improper solution space could be a huge burden for computation speed. Some comparisons with some classic planning methods may be helpful.
2. How to define the feasibility of solution space? Because there are always some detrimental demonstrations existing. Maybe a not suitable space can cause an undesirable outcome. The authors can make this part clearer.
3. For a Post-hoc solution search (Fig. 10), a set of trajectories close to the demonstration are obtained, but the relative extent is not discussed. A comparison between similar methods can make your contributions outstanding.
4. The authors proposed a method to find a potential solution from demonstrations. However, the authors should elaborate the application values. For example, learning from demonstrations in auto-driving technologies usually serves for human-like driving or driving safety. However, the treatment proposed in this study seemingly discusses a new approach without telling us what its meaning is. Following the descriptions of this study, the advantage is to find a solution through learning from demonstrations, but there have been many ways that can complete this. Maybe application potential and corresponding validations can help to make your contributions more valuable.
5. Reproducibility is unknown because the codes are not open.

---

> ### Author Response · Authors · 2022-11-19
> **Authors's response (2/2)**
>
>
>
> ***“How about planning efficiency? Planning time consumption is vital to driving safety, and generally, an improper solution space could be a huge burden for computation speed. Some comparisons with some classic planning methods may be helpful.”***
>
>
>
> We would like to clarify that planning in our context does not mean “real-time” planning, but rather the selection of one plan or solution (i.e., a path for 2d planning and a policy $\pi(a\vert s)$ for highway driving) out of many optimised plans/solutions for a given task after building the solution archive. Where each plan presents a unique behaviour. We select the plan at the beginning of the task execution and stick to the plan as we assume a static environment. This is suitable for tasks like pushing objects, picking and placing objects and so on (with different behaviours) where we do not have to adapt our plan in real-time. However, in the case of highway driving, as we have a step-wise controller, it is possible to select different policies to be executed at different time instances if needed.
>
>
> ***“ Reproducibility is unknown because the codes are not open”***
>
> A discussion about reproducibility is included in Appendix A.3, where all the hyperparameters and the environment setups are listed. We run all the experiments several times to ensure that our approach converges to the same solution space. In Appendix A.5, we show the mean solution sizes and mean valid solutions sizes, where indeed, the low standard deviation shows that the final dimension of the solution spaces is consistent during the optimisation. The code will be released publicly in the future.

---

> ### Author Response · Authors · 2022-11-19
> **Authors' response (1/2)**
>
>
> ***[Summary of paper] “This study built a parameterized solution space to help model driving behaviours for planning, which considering the effect of individual differences of driving demonstrations.”***
>
>
> Our approach does work for both planning (path planning and robotic arm movement planning) and control (driving environment with a step-wise controller).
>
>
> ***“How to define the feasibility of solution space? Because there are always some detrimental demonstrations existing. Maybe a not suitable space can cause an undesirable outcome. The authors can make this part clearer.”***
>
> - The feasibility of the solutions is verified by the simulator. As pointed out by the reviewer, the provided demonstrations can be detrimental. This scenario is considered with the robotic arm manipulation experiment. Here, the demonstrations were recorded using a motion capture system, capturing only the motion of the object being pushed by a human on the table. The motions are not actually executed (and recorded) with the robot. As a result, some demonstrations are actually infeasible for the robot due to its physical limitations (i.e., joint limits). However, during policy optimization, as the solutions are generated with access to a high-fidelity simulator of the robot, the optimizer rules out the non-feasible behaviours using the feedback received from the simulator.
>
> - Internally, we did consider a scenario where the solutions are generated in a low fidelity simulator, i.e., using the toy point environment instead of the Franka arm with the demonstrations intended for Franka Arm. As suggested by the reviewer, the generated solutions would ignore the physical limitation of the robot as the toy environment does not provide any physical constraint feedback to the agent, i.e., collision detection of 3D obstacles with all the joints of the robot. However, the resulting solution archive still contains solutions that satisfy the actual constraints (i.e., by filtering out the resulting solutions by running them on the simulator). This is an example of the post-hoc policy search that finds policies that satisfy different environments' constraints. This is now included in Appendix 6.
>
>
> ***“For a Post-hoc solution search (Fig. 10), a set of trajectories close to the demonstration are obtained, but the relative extent is not discussed. A comparison between similar methods can make your contributions outstanding.”***
>
>
> Thank you for pointing it out! That is actually a very good idea. We’ve included in the updated paper a policy behaviour matching comparison using a converged VAE and our method (see Fig 10). As our projection function provides a (statistical) summary of the state distributions for the trajectories, our results show that our approach has a better behaviour matching mechanism.
>
>
> ***“The authors proposed a method to find a potential solution from demonstrations. The authors should elaborate the application values....However, the treatment proposed in this study seemingly discusses a new approach without telling us what its meaning is. ... the advantage is to find a solution through learning from demonstrations... corresponding validations can help to make your contributions more valuable.”***
>
>
>
> - Thank you for pointing it out! It is true that we are proposing a new framework that is essentially imitation-guided RL to generate multimodal behaviours beyond the expert demonstrations, we have updated the introduction with clear intuition and the contributions are highlighted.
>
> - We have included in the updated version also a new appendix section (A.8) discussing the potential applications of our approach. To summarise, one potential application can be data augmentation that augments the original demonstrations dataset with the generated behaviours to train other networks, such as shown in the example of VAE (Fig. 7). For example, we can see that VAE fails in dealing with a small dataset while it can be fully trained using data generated from our approach. Furthermore, take the example of Franka Arm with a real physic engine and 3D simulation. We can collect visual demonstrations using policies from our solution archive, which can be used to train other vision-input based networks that typically require a large set of input data to guarantee convergence. Lastly, our method deals with states-only observations, and it can generate policies that predict actions to produce such observations.

---

### Official Review · Reviewer_Vtfi · 2022-10-31

**Confidence:** 5
**Correctness:** 2
**Technical Novelty And Significance:** 2
**Empirical Novelty And Significance:** 2
**Recommendation:** 3

**Clarity, Quality, Novelty And Reproducibility:**

The paper is well-written and easy to follow. Novelty is limited. Experimental set-ups are too simple to understand the limitations of the approach.

**Strength And Weaknesses:**

- Learning from demonstrations is an active area of research, and of interest to the wider community. Encoding and generating diverse behaviors from demonstrations are important from both what to imitate and how to imitate perspective.

- Problem formulation of the paper is similar to apprenticeship learning (or inverse reinforcement learning) under different preferences of the expert. In the problem formulation, the reward function needs to be parameterized in the preferences that the expert demonstrations are looking to encode. Eq.3 objective of matching state-visitation frequency under different preferences of the expert has also been well-studied in discrete spaces. Moreover, ,ean and mode seeking LfD is well-known in literature.

- In continuous spaces, the hidden markov model/hidden semi-markov model is better suited to encode the temporal sequence in the state observations, as compared to the GMM (see [1]). Subsequently, Viterbi decoding can be used for generating diverse demonstrations.

- There are no comparisons with other behavior cloning approaches to establish the merits of the approach. Moreover, there is no numerical quantification between expert demonstrations and generated demonstrations to quantify the performance.

[1] Sequential robot imitation learning from observations, IJRR, 2021


**Summary Of The Paper:**

This paper presents a learning from demonstration approach to generate diverse policies representing the different preferences of the expert. The authors seek to learn a GMM in the latent space, and match the state-visitation frequency from state observations only. Experiments on simulated reaching tasks with robot arm and car driving suggest the feasibility of the approach.

**Summary Of The Review:**

This paper presents a learning from demonstration approach to encode and decode diverse expert demonstrations. The problem is of interest to a wider community. Consolidating the problem formulation further, quantifying the performance in the experiments and comparison with other approaches will improve the quality of the paper.

---

> ### Author Response · Authors · 2022-11-19
> **Authors's response (2/2)**
>
>
>
> ***“Moreover, there is no numerical quantification between expert demonstrations and generated demonstrations to quantify the performance.”***
>
>
> We apologize for not having an explicit reference for the numerical quantification. We do have Appendix A.5, which was under a different name, intended for this. We’ve changed its title now to ”ABLATION STUDIES AND NUMERICAL QUANTIFICATION OF SOLUTION ARCHIVE“  to make this information clear. As our goal is to discover all the possible policies that maximise a given task reward function under different constraints, the performance is measured by the number of total solutions generated by our solution archive with respect to the number of expert demonstrations. In addition, we also include as a performance indicator, the total number of valid solutions (i.e., satisfying the task constraints as reaching the goal and avoiding the obstacles). We further discuss the quality of the solution archives under different setups for more ablation study, such as the choice of encoding, choice of GMM in affecting the quality of the solution archives, etc.
>
>
> ***“This paper presents a learning from demonstration approach to encode and decode diverse expert demonstrations.”***
>
> This statement is incorrect. We are proposing an imitation-guided RL approach to discover diverse behaviours beyond the provided expert state-only demonstrations without labels. We've updated the paper to make this point clear. Furthermore, we do not have any decoding mechanism for the generation of demonstrations or decoding experts’ demonstrations. Our encoding mechanism is used to project the policies into a latent (preference) space according to their state distributions.

---

> > ### Comment · Reviewer_Vtfi · 2022-11-30
> > **response to authors**
> >
> > It seems to me that the problem formulation is similar to multi-task apprenticeship learning setup where the goal is to recover policies that match different expert preferences. This objective of the paper on learning multiple strategies under different expert preference is not novel.
> >
> > Thanks for correcting my statement on state-visitation to state-region visitation frequency, and providing other detailed feedback.
> >
> > I believe the paper needs more work in problem formulation as different preferences of the expert need to be parameterized to be able to recover those policies explicitly. Matching state-region frequencies is a plausible way, and would benefit from comparison with other methods to improve the quality of the work.

---

> ### Author Response · Authors · 2022-11-19
> **Authors' response (1/2)**
>
> ***[Summary of paper] “Authors seek to learn a GMM in the latent space and match the state-visitation frequency from state observations only.***
>
> It is incorrect to say that we learn GMM in the latent space. Instead, we use the GMM fitted to the given state-only demonstrations to define a latent preference space (see Sec.3), where each mixture component represents a state region. Then the preference is defined as different state-region visitation frequencies in a demonstration, which is captured by our proposed projection function as low dimensional encoding.
> As we deal with state-regions and each state region is modelled by a mixture component (i.e., Gaussian distribution), we are matching the underlying state distributions rather than individual state-visitations.
>
> ***“formulation of the paper is similar to apprenticeship learning (or inverse reinforcement learning) under different preferences of the expert.”***
>
> This statement is incorrect, as we are formulating the problem as a constrained optimisation problem (see Sec. 3 and Eq 1 and 2). Unlike the AL or IRL/IL, we *do not try to purely mimic the given demonstrations*. Instead, we generate multimodal behaviours that satisfy different preference constraints, which go beyond the provided (state-only) demonstrations. We have updated the introduction and the contribution section to explain these points more clearly.
>
> ***. Eq.3 objective of matching state-visitation frequency under different preferences of the expert has also been well-studied in discrete spaces. Moreover, mean and mode seeking LfD is well-known in literature.***
>
> - Eq.3 does not match the state-visitation frequency, but the state-region visitation represented by the encoding. More details can be found in Sec. 3. Given the multimodal demonstrations, GMM models each state region from the state-observation with one mixture component (see Fig. 3). Where state-region visitation is given by the encoding function (see Sec. 3). And as pointed out by the reviewer, the state-visitation frequency matching is only applicable for discrete space, while our approach is valid for both discrete and continuous spaces.
>
> - We agree that mean and mode-seeking LfD are well-known in the literature. And we intend to avoid our method exhibiting either mean or mode-seeking behaviours, but instead, retain all the possible solution modes and, at the same time, discover more solution modes, i.e., solution modes as results of interpolations from the existing modes.
>
> ***“In continuous spaces, the hidden Markov model/hidden semi-Markov model is better suited to encode the temporal sequence in the state observations, as compared to the GMM (see [1]). Subsequently, Viterbi decoding can be used for generating diverse demonstrations."***
>
>
> We agree that an HMM is a possible alternative to the proposed GMM model. In our formulation, the transition probability between different mixture components is not considered as the preference, and it is defined only by the state-region visitation. Thus, we did not include any transition information in our encoding. The use of the Viterbi algorithm might not be straightforward for our approach, as it seeks the most likely trajectories. While we intend to generate also novel trajectories that might have low likely transitions. The use of HMM and the consideration of transition as additional preference can be certainly proposed as future work. We've included a small discussion to this in the updated version and cited the reference paper in Appx. A.5.1
>
>
>
>
> ***“There are no comparisons with other behavior cloning approaches to establish the merits of the approach“***
>
>
>
> - We did provide the theoretical analysis (see Sec 2.1 and Appendix A.2) about why standard imitation learning approaches (BC, Adversarial IL, etc) would not address the issues of multimodal demonstrations. For the path planning experiment, we compare the result of VAE both in dealing with the generation of trajectories and encoding capabilities (Fig. 7). Results from both GMR and GAN are also provided in the appendix (Fig. 11 and Fig. 12) as stochastic BC approach and generative model for encoding/behaviour generation. As GAN-based methods do not perform well with a small dataset, which results from mode-collapse (see discussion 2.1 and Appx A.2 for more details for Adversarial IL/IRL) and unstable convergence, we trained GAN using solutions generated from our method. Thus, we show the usefulness of our method in providing more training data for other approaches.
>
> - In addition, existing IL methods do not allow us to perform the post-hoc solution search or policy matching from demonstration, as we proposed in our method.

---

### Author Response · Authors · 2022-11-18
**General response**


We would like to thank all reviewers for their valuable time and dedication to our paper. We have responded to the concerns raised by the reviewers and updated the relevant sections in the paper. More specifically, we have clarified the main concerns such as the justification for the use of the simulator, interpretability of the encoding, reproducibility of the approach, and comparison to similar methods to the respective reviewers.

The update of the paper includes minor modifications to the Abstract and the Introduction section to clarify and highlight our contribution, and several Appendix sections which show more experiments and ablation study results analysis and the applicability of our method.

To clarify, we proposed a new imitation-guided RL framework that combines RL with Imitation Learning to generate a diverse range of behaviours with limited **unlabelled** “state-only demonstrations”. Unlike most of the Learning from Demonstration (LfD) and Imitation Learning from Observation Only (ILFO) methods, our goal is not just to learn *how to perform a task* (either with mean-seeking or mode-seeking policy) nor *how to imitate the demonstrators*, but rather *how to perform a task in all possible ways* (multi-modal policy) beyond the demonstrated data. Thus, we focus on applications where a high-level task reward function can be defined intuitively; however, the reward for the preference component that causes diverse behaviour cannot be explicitly defined. We hypothesise that the expert policies are the results of solving a constrained policy optimisation problem, where the policies are optimised by maximising the common task reward function while constraining their behaviours to follow certain behaviour patterns. These constraints are defined as different individual preferences. We define the preferences as different state distributions. The generation of multi-modal behaviours results from optimising policies for the task reward with different preference instances. Moreover, the use of the simulator is required under state-only demonstrations setup, where supervised or unsupervised learning approaches cannot be applied directly to find a policy, and the agent must interact with the environment or with a simulator [1].

[1] Sun, Wen, et al. "Provably efficient imitation learning from observation alone." International conference on machine learning. PMLR, 2019.

---

### Decision · Program_Chairs · 2023-01-20

**Decision:**

Reject

**Justification For Why Not Higher Score:**

Main argument against the paper: the use of GMMs leads to a non-scaling solution.

**Justification For Why Not Lower Score:**

n/a

**Metareview: Summary, Strengths And Weaknesses:**

Modeling observations with GMMs, this paper uses those to create a latent space, which is then used to create policies with MAP-ELITES.

The methods proposed in the paper are incremental, and not supported by a comparative experimental section.  Furthermore, the experiments are all for low-dimensional systems---a necessary side effect of the use of GMMs.

Applicability of the proposal method over existing ones is not clarified.